# HANDLOOM: Learned Tracing of One-Dimensional Objects for Inspection and Manipulation

**Vainavi Viswanath\*[1], Kaushik Shivakumar\*[1], Mallika Parulekar[†1], Jainil Ajmera[†1],**
**Justin Kerr[1], Jeffrey Ichnowski[2], Richard Cheng[3], Thomas Kollar[3],**
**Ken Goldberg[1]**
\* equal contribution, † equal contribution
`vainaviv@berkeley.edu, kaushiks@berkeley.edu`

**Abstract:** Tracing – estimating the spatial state of – long deformable linear objects such as cables, threads, hoses, or ropes, is useful for a broad range of tasks in homes, retail, factories, construction, transportation, and healthcare. For long deformable linear objects (DLOs or simply cables) with many (over 25) crossings, we present HANDLOOM (Heterogeneous Autoregressive Learned Deformable Linear Object Observation and Manipulation) a learning-based algorithm that fits a trace to a greyscale image of cables. We evaluate HANDLOOM on semi-planar DLO configurations where each crossing involves at most 2 segments. HANDLOOM makes use of neural networks trained with 30,000 simulated examples and 568 real examples to autoregressively estimate traces of cables and classify crossings. Experiments find that in settings with multiple identical cables, HANDLOOM can trace each cable with $80\%$ accuracy. In single-cable images, HANDLOOM can trace and identify knots with $77\%$ accuracy. When HANDLOOM is incorporated into a bimanual robot system, it enables state-based imitation of knot tying with 80% accuracy, and it successfully untangles $64\%$ of cable configurations across 3 levels of difficulty. Additionally, HANDLOOM demonstrates generalization to knot types and materials (rubber, cloth rope) not present in the training dataset with 85% accuracy. Supplementary material, including all code and an annotated dataset of RGB-D images of cables along with ground-truth traces, is at https://sites.google.com/view/cable-tracing.

**Keywords:** state estimation, deformable manipulation

## 1 Introduction

Tracing long one-dimensional objects such as cables has many applications in robotics. However, this is a challenging task as depth images are prone to noise, and estimating cable state from greyscale images alone is difficult since cables often fall into complex configurations with many crossings. Long cables can also contain a significant amount of free cable (*slack*), which can occlude and inhibit the perception of knots and crossings.

Prior cable tracing research use a combination of learned and analytic methods to trace a cable, but are limited to at most 3 crossings [1, 2]. Previous cable manipulation work bypasses state estimation with object detection and keypoint selection networks for task-specific points based on geometric patterns [3, 4, 5, 6, 7, 8]. Sundaresan et al. [9] employ dense descriptors for cable state estimation in knot tying but focus on thick short cables with loose overhand knots. However, this work tackles the challenges of long cables in semi-planar configurations (at most 2 cable segments per crossing), which often contain dense arrangements with over 25 crossings and near-parallel cable segments. Although works like Lui and Saxena [10] and Huang et al. [11] also perform cable state estimation,

---

[1]University of California, Berkeley. [2]Carnegie Mellon University. [3]Toyota Research Institute.

7th Conference on Robot Learning (CoRL 2023), Atlanta, USA.

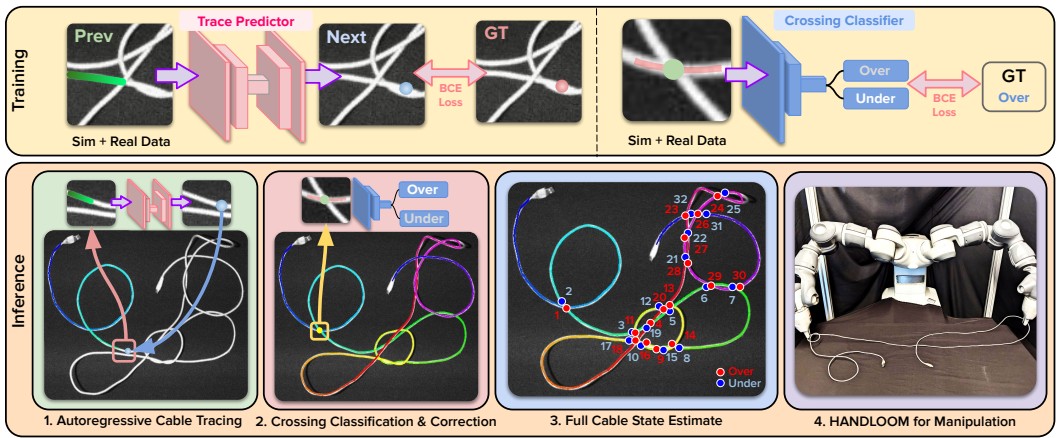

Figure 1: **Example of HANDLOOM on a single knotted cable with 16 crossings:** HANDLOOM includes two networks. One network (pink at the top) is trained to predict the next point in the trace given a prior context window, and the other network (blue at the top) is trained to classify over- and under-crossings. During inference, HANDLOOM 1) uses the pink network to autoregressively find the most likely trace (illustrated with a rainbow gradient, from violet to purple, depicting the path of the cable) and 2) performs crossing recognition using the blue network to obtain the full state of the cable (3), where red circles indicate overcrossings and blue circles indicate undercrossings. 4) The state estimate from HANDLOOM can be used for inspection and manipulation.

they use analytic methods and do not address the challenging cases mentioned above which we consider. While analytic methods require heuristics to score and select from traces, learning to predict cable traces directly from images avoids this problem.

This work considers long cables (up to 3 meters in length) in semi-planar configurations, i.e. where each crossing includes at most 2 cable segments. These cable configurations may include knots within a single cable (e.g. overhand, bowline, etc.) or between multiple cables (e.g. carrick bend, sheet bend, etc.). This paper contributes:

1. Heterogeneous Autoregressive Learned Deformable Linear Object Observation and Manipulation (HANDLOOM), shown in Figure 1: A tracing algorithm for long deformable linear objects with up to 25 or more crossings.

2. Novel methods for cable inspection, state-based imitation, knot detection, and autonomous untangling for cables using HANDLOOM.

3. Data from physical robot experiments that suggest HANDLOOM can correctly trace long DLOs unseen during training with 85% accuracy, trace and segment a single cable in multi-cable settings with 80% accuracy, and detect knots with 77% accuracy. Robot experiments suggest HANDLOOM in a physical system for untangling semi-planar knots achieves 64% untangling success in under 8 minutes and in learning from demonstrations, achieves 80% success.

4. Publicly available code and data (and data generation code) for HANDLOOM: https://github.com/vainaviv/handloom.

## 2 Related Work

### 2.1 Deformable Object Manipulation

Recent advancements in deformable manipulation include cable untangling algorithms [5, 6, 7, 10], fabric smoothing and folding techniques [12, 13, 14, 15, 16, 17, 18], and object placement in bags [19, 20]. Methods for autonomously manipulating deformable objects range from model-free to model-based, the latter of which estimates the state of the object for subsequent planning.

**Model-free approaches** include reinforcement or self-supervised learning for fabric smoothing and folding [21, 22, 23, 24] and straightening curved ropes [22], or directly imitating human actions [12, 25]. Research by Grannen et al. [6] and Sundaresan et al. [7] employs learning-based keypoint detection to untangle isolated knots without state estimation. Viswanath et al. [3] and Shivakumar et al. [4] extend this approach to long cables (3m) with a learned knot detection pipeline. However, scaling to arbitrary knot types requires impractical amounts of human labels across many conceivable knot types. Our study focuses on state estimation to address this limitation.

**Model-based methods** for deformable objects employ methods to estimate the state as well as dynamics. Work on cable manipulation includes that by Sundaresan et al. [9], who use dense descriptors for goal-conditioned manipulation. Descriptors have also been applied to fabric smoothing [14], as well as visual dynamics models for non-knotted cables [26, 27] and fabric [15, 26, 28]. Other approaches include learning visual models for manipulation [29], iterative refinement of dynamic actions [30], and using approximate state dynamics with a learned error function [31]. Fusing point clouds across time has shown success in tracking segments of cable provided they are not tangled on themselves [32, 33]. Works from Lui and Saxena [10] and Huang et al. [11] estimate splines of cables before untangling them, and we discuss these methods among others below.

## 2.2 Tracing Deformable Linear Objects

Prior tracing methods for LDOs, including in our prior work [4], primarily employ analytic approaches [2, 10, 11, 34]. Other works, Jackson et al. [35] and Padoy and Hager [36], optimize splines to trace surgical threads. Very recently, Kicki et al. [2] use a primarily analytic method for fast state estimation and tracking of short cable segments with 0-1 crossings. In contrast to these works, this work focuses on longer cables with a greater variety of configurations including those with over 25 crossings and twisted cable segments. Other prior work attempts to identify the locations of crossings [37] in cluttered scenes of cables but does not fully estimate cable state.

Several prior works approach sub-parts of the problem using learning. Lui and Saxena [10] use learning to identify weights on different criteria to score traces. Huang et al. [11] focus on classifying crossings for thick cables. Song et al. [8] predict entire traces for short cables as gradient maps but lack quantitative results for cables and evaluation of long cables with tight knots. Yan et al. [38] use self-supervised learning to iteratively estimate splines in a coarse-to-fine manner. In very recent work, Caporali et al. [1, 39] use learned embeddings to match cable segments on opposite sides of analytically identified crossings, in scenes with 3 or fewer total crossings, while we test HANDLOOM on complex configurations containing over 25 crossings.

## 3 Problem Statement

**Workspace and Assumptions:** The workspace is defined by an $(x, y, z)$ coordinate system with a fixed overhead camera facing the surface that outputs grayscale images. We assume that 1) the greyscale image includes only cables (no obstacles), 2) each cable is visually distinguishable from the background, 3) each cable has at least one endpoint visible, and 4) the configuration is semiplanar, meaning each crossing contains at most 2 cable segments. We define each cable state for $i = 1, ..., l$ cables to be $\theta_i(s) = \{(x(s), y(s), z(s))\}$ where $s$ is an arc-length parameter that ranges $[0, 1]$, representing the normalized length of the cable. Here, $(x(s), y(s), z(s))$ is the location of a cable point at a normalized arc length of $s$ from the cable's first endpoint. We also define the range of $\theta(s)$—that is, the set of all points on a cable at time $t$—to be $\mathcal{C}_t$. We assume all cables are visually distinguishable from the background and that the background is monochrome. Additional assumptions we make for the manipulation tasks are stated in Section 5.4.2.

**Objective:** The objective of HANDLOOM is to estimate the cable state – a pixel-wise trace as a function of $s$ of one specified cable indicating all over- and under-crossings.

## 4 Algorithm

HANDLOOM (Figure 1) includes a learned cable tracer that estimates the cable's path through the image and a crossing classifier with a correction method that refines predictions.

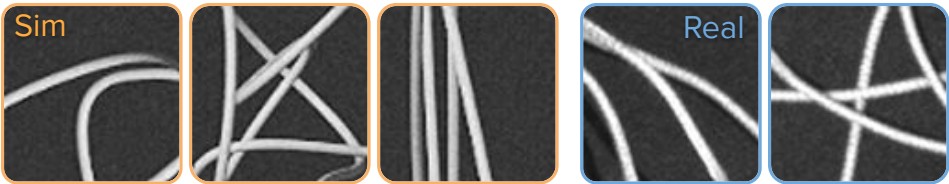

Figure 2: **Cropped images of simulated and real cables for training HANDLOOM**: On the left are simulated cable crops augmented with Gaussian noise, brightness, and sharpening to match the real images (right).

## 4.1 Learned Cable Tracer Model

We break the problem of cable tracing down into steps, where each step generates a probability distribution for the next point based on prior points. To achieve this, we employ a learned model that takes an image crop and trace points from previous iterations and predicts a heatmap representing the probability distribution of the next pixel's location. By operating on local information within crops, the model mitigates overfitting and facilitates sim-to-real transfer, focusing on local characteristics rather than global visual and geometric attributes like knots.

More formally, representing the grayscale image as $I$, and each trace $s_{i,\text{tot}}$ in the image as a sequence of pixels $s_{i,0}, s_{i,1}, ... s_{i,n}$, we break the probability distribution over traces conditioned on the image into smaller, tractable pieces using the chain rule of probability. $f_\theta$ is a learned neural network, $\text{crop}(I, p)$ is a crop of image $I$ centered at pixel $p$, and $k$ is the context length.

$$P(s_{i,\text{tot}}|s_{i,0}, I) = \prod_{j=1}^{n} P(s_{i,j}|s_{i,0}...s_{i,j-1}, I) \approx \prod_{j=1}^{n} f_\theta(s_{i,j}|s_{i,j-k}...s_{i,j-1}, \text{crop}(I, s_{i,j-1}))$$

### 4.1.1 Dataset and Model Training

To train the crossing classifier, we simulate a diverse range of crossing configurations to generate a dataset. We use Blender [40] to create 30,000 simulated grayscale images that closely resemble real observations (Fig. 2). Cable configurations are produced through three methods with random Bezier curves: (1) selecting points outside a small exclusion radius around the current point, (2) intentionally creating near-parallel segments, and (3) constraining specific cable segments to achieve a dense and knot-like appearance in certain spatial regions. The curves are colored white and have slightly randomized thicknesses.

We randomly sample image crops along the cable of interest, with a focus on cable crossings (representing $95\%$ of samples). The simulated images are augmented with pixel-wise Gaussian noise with standard deviation of 6, brightness with standard deviation of 5, and sharpening to imitate the appearance of real cables. Additionally, we include a smaller dataset of 568 hand-labeled real cable crop images, sampled such that it comprises approximately 20% of the training examples. Training employs the Adam optimizer [41] with pixelwise binary cross-entropy loss, using a batch size of 64 and a learning rate of $10^{-5}$.

### 4.1.2 Model Architecture and Inference

We use the UNet architecture [42]. We choose trace points spaced approximately 12 pixels apart, chosen by grid search, balancing between adding context and reducing overfitting. We further balance context and overfitting by tuning the crop size ($64 \times 64$) and the number of previous points fed into the model (3) through grid search.

Naively training the model for cable tracing in all possible initial directions leads to poor performance, as it would require learning rotational equivariance from data, reducing data efficiency. To overcome this, we pre-rotate the input image, aligning the last two trace points horizontally and ensuring the trace always moves left to right, optimizing the model's capacity for predicting the next trace point. We then rotate the output heatmap back to the original orientation.

During inference, initializing the trace requires an input of a single start pixel along the cable (in practice, one endpoint). We use an analytic tracer as in [4] to trace approximately 4 trace points and use these points to initialize the learned tracer, which requires multiple previous trace points to predict the next point along the trace.

The tracer autoregressively applies the learned model to extend the trace. The network receives a cropped overhead image centered on the last predicted trace point ($64 \times 64$ pixels). Previous trace points are fused into a gradient line (shown in Figure 1), forming one channel of the input image. The other two channels contain an identical grayscale image. The model outputs a heatmap ($64 \times 64 \times 1$) indicating the likelihood of each pixel being the next trace point. We greedily select the argmax of this heatmap as the next point in the trace. This process continues until leaving the visible workspace or reaching an endpoint, which is known using similar learned detectors as Shivakumar et al. [4].

### 4.2 Over/Undercrossing Predictor

#### 4.2.1 Data and Model Input

We use $20 \times 20$ simulated and real over/undercrossing crops. The 568 real images are oversampled such that they are seen 20% of the time during training. Augmentation methods applied to the cable tracer are also used here to mimic the appearance of real cable crops. The network receives a $20 \times 20 \times 3$ input crop. The first channel encodes trace points fused into a line, representing the segment of interest. To exploit rotational invariance, the crop is rotated to ensure the segment's first and last points are horizontal. The second channel consists of a Gaussian heatmap centered at the target crossing position, providing positional information to handle dense configurations. The third channel encodes the grayscale image of the crop.

#### 4.2.2 Model Architecture and Inference

We use a ResNet-34 classification model with a sigmoid activation to predict scores between 0 and 1. The model is trained using binary cross-entropy loss. We determine the binary classification using a threshold of 0.275 (explanation for this number is in the appendix Section 7.1.1). The algorithm uses the fact that each crossing is encountered twice to correct errors in the classifier's predictions, favoring the higher confidence detection and updating the probability of the crossing to $(1-$ the original value), storing the confidences for subsequent tasks. The learned cable tracer, over/undercrossing predictor, and crossing correction method combined together result in a full cable state estimator: HANDLOOM.

### 4.3 Using HANDLOOM in Downstream Applications

**Tracing in Multi-cable Settings:** We apply HANDLOOM to inspection in multi-cable settings for tasks like locating the power adapter of a cable tangled with other visually similar cables given the endpoint. HANDLOOM is fed an endpoint and returns a trace to the adapter connected to it.

**Learning from Demonstrations:** We use HANDLOOM to enable physical, state-based knot tying in the presence of distractor cables, which is much more challenging for a policy operating on RGB observations rather than underlying state. Demonstrations are pick-place actions, parameterized by absolute arc-length or arc-length relative to crossings. More details are in Appendix Section 7.3.

**Robot Cable Untangling:** For cable untangling, HANDLOOM is combined with analytic knot detection, untangling point detection techniques, and bi-manual robot manipulation primitives to create a system for robot untangling. Appendix Section 7.2 contains more details.

## 5 Physical Experiments

We evaluate HANDLOOM on 1) tracing cables unseen during training, 2) cable inspection in multi-cable settings, 3) learning knot tying from demonstrations, 4) knot detection, and 5) untangling.

The workspace has a 1 m $\times$ 0.75 m foam-padded, black surface, a bimanual ABB YuMi robot, and an overhead Photoneo PhoXi camera with $773 \times 1032 \times 4$ RGB-D observations. HANDLOOM is fed only the RGB image, but the depth data is used for grasping. The PhoXi outputs the same values

Table 1: Generalization of HANDLOOM to Different Cable Types

| Cable Reference | Length (m) | Color | Texture | Physical Properties | HANDLOOM Succ. Rate |
|---|---|---|---|---|---|
| TR (trained with) | 2.74 | White/gray | Braided | Slightly stiff | 6/8 |
| 1 | 2.09 | Gray | Rubbery | Slightly thicker than TR | 7/8 |
| 2 | 4.68 | Yellow with black text | Rubbery and plastic | Very stiff | 8/8 |
| 3 | 2.08 | Tan | Rubbery | Highly elastic | 7/8 |
| 4 | 1.79 | Bright red | Braided | Flimsy | 6/8 |
| 5 | 4.61 | White | Braided | Flimsy | 6/8 |

across all 3, making the observations grayscale and depth. See Appendix, Section 7.4 for details on failure modes for each experiment.

## 5.1 Using HANDLOOM for Tracing Cables Unseen During Training

For this perception experiment, the workspace contains a single cable with one of the following knots: overhand, figure-eight, overhand honda, or bowline. Here, we provide HANDLOOM with the two endpoints to test the tracer in isolation, independent of endpoint detection. We report a success if the progression of the trace correctly follows the path of the cable without deviating.

Results in Table 1 show HANDLOOM can generalize to cables with varying appearances, textures, lengths, and physical properties. Cables can also be seen in Appendix Figure 9. HANDLOOM performs comparably on cable TR (the cable it was trained with) as it does on the other cables.

## 5.2 Using HANDLOOM for Cable Inspection in Multi-Cable Settings

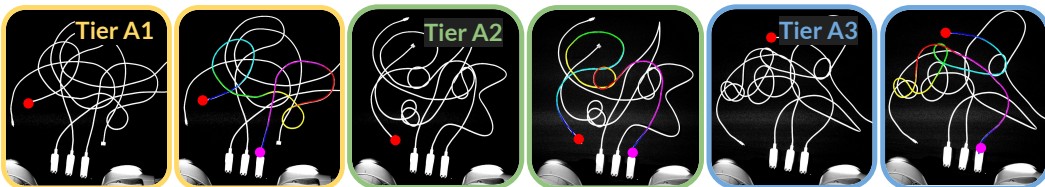

Figure 3: **Multi-cable tracing**: here are 3 pairs of images. The left of each pair illustrates an example from the tier and the right is the successful trace. The traces from left to right encounter 24, 13, and 28 crossings.

For cable inspection, the workspace contains a power strip. Attached to the power strip are three white MacBook adapters with two 3 m USB C-to-C cables and one 2 m USB-C to MagSafe 3 cable (shown in Figure 3). The goal is to provide the trace of a cable and identify the relevant adaptor given the endpoint. We evaluate perception in multi-cable settings across 3 tiers of difficulty.

1. **Tier A1**: No knots; cables are dropped onto the workspace, one at a time.
2. **Tier A2**: Each cable is tied with a single knot (figure-eight, overhand, overhand honda, bowline, linked overhand, or figure-eight honda) measuring 5-10 cm in diameter, and subsequently dropped onto the workspace one by one.
3. **Tier A3**: Similar to tier A2 but contains the following 2-cable knots (square, carrick bend, and sheet bend) with up to three knots in the scene.

Across all 3 tiers, we assume the cable of interest cannot exit and re-enter the workspace and that crossings must be semi-planar. Additionally, we pass in the locations of all three adapters to the tracer and an endpoint to start tracing from. To account for noise in the input images, we take 3 images of each configuration. We count a success if a majority (2 of the 3 images) have the correct trace (reaching their corresponding adapter); otherwise, we report a failure.

We compare the performance of the learned tracer from HANDLOOM against an analytic tracer from Shivakumar et al. [4] as a baseline, using scoring rules inspired by Lui and Saxena [10] and Keipour et al. [34]. The analytic tracer explores potential paths and selects the most likely trace based on a scoring metric [4], prioritizing paths that reach an endpoint, have minimal angle deviations, and have high coverage scores. Table 2 shows that the learned tracer significantly outperforms the baseline analytic tracer on all 3 tiers of difficulty with a total of 80% success across the tiers.

## 5.3 Using HANDLOOM for Physical Robot Knot Tying from Demonstrations

Knot-tying demonstrations are conducted on a nylon rope of length 147 cm and a diameter of 7 mm. We tune our demonstrations to work on plain backgrounds; however, during rollouts, we add distractor cables that intersect the cable to be tied. These distractor cables are of identical types to the cable of interest; thus, manipulating the correct points requires accurate cable state estimation from HANDLOOM. Although these cables are thicker, more twisted, and less stiff than the cable HANDLOOM is trained with, HANDLOOM is able to generalize, tracing the cable successfully in 13 out of 15 cases. We evaluate the policy executed by the YuMi bimanual robot on the following: **Tier B1:** 0 distractors, **Tier B2:** 1 distractor, and **Tier B3:** 2 distractors. We count a success when a knot has been tied (i.e. lifting the endpoints results in a knot's presence).

Table 4 show 86% tracing success and 80% robotic knot tying success across the 3 tiers, suggesting HANDLOOM can apply policies learned from real world demonstrations, even with distractors.

## 5.4 Using HANDLOOM for Knot Detection and Physical Robot Untangling

To test HANDLOOM applied to the knot detection and physical untangling task, we use a single 3 m white, braided USB-A to micro-USB cable.

### 5.4.1 Knot Detection

The details on the state-based knot detection method, which uses the sequence of over- and under-crossings to identify knots, can be found in the Appendix Section 7.2.2.

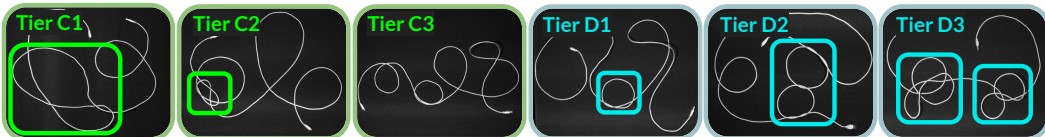

Figure 4: **Starting configurations** for the tiers for HANDLOOM experiments and robot untangling experiments. Here is the crossing count for these examples from left to right: 5, 10, 6, 4, 9, and 14.

We evaluate HANDLOOM on 3 tiers of cable configurations, shown in Figure 4. The ordering of the categories for these experiments does not indicate varying difficulty. Rather, they are 3 categories of knot configurations to test HANDLOOM on.

1. **Tier C1**: Loose (35-40 cm in diameter) figure-eight, overhand, overhand honda, bowline, linked overhand, and figure-eight honda knots.

2. **Tier C2**: Dense (5-10 cm in diameter) figure-eight, overhand, overhand honda, bowline, linked overhand, and figure-eight honda knots.

3. **Tier C3**: Fake knots (trivial configurations positioned to appear knot-like from afar).

We evaluate HANDLOOM on the following 3 baselines on the 3 tiers.

1. SGTM 2.0 [4] perception system: using a Mask R-CNN model trained on overhand and figure-eight knots for knot detection.

2. HANDLOOM (-LT): replacing the Learned Tracer with the same analytic tracer from Shivakumar et al. [4] as described in Section 5.2 combined with the crossing identification.

3. HANDLOOM (-CC): using the learned tracer and crossing identification scheme to do knot detection without Crossing Cancellation, covered in the appendix (Section 7.2.3).

We report the success rate of each of these algorithms as follows: if a knot is present, the algorithm is successful if it correctly detects the first knot (i.e. labels its first undercrossing); if there are no knots, the algorithm is successful if it correctly detects that there are no knots.

As summarized in Table 3, knot detection using HANDLOOM considerably outperforms SGTM 2.0, HANDLOOM (-LT), and HANDLOOM (-CC) on tiers C1 and C3. SGTM 2.0 marginally outperforms HANDLOOM in tier C2. This is because SGTM 2.0's Mask R-CNN is trained on dense overhand and figure-eight knots, which are visually similar to the knots in tier C2.

Table 2: Multi-Cable Tracing

| Tier | Analytic | Learned |
|------|----------|---------|
| A1 | 3/30 | **26/30** |
| A2 | 2/30 | **23/30** |
| A3 | 1/30 | **23/30** |

Learned (HANDLOOM) compared to an
analytic tracer from Shivakumar et al. [4].

Table 3: Knot Detection Experiments

| Tier | SGTM 2.0 | HL (-LT) | HL (-CC) | HL |
|------|----------|----------|----------|-----|
| C1 | 2/30 | 14/30 | 20/30 | **24/30** |
| C2 | **28/30** | 8/30 | 21/30 | 26/30 |
| C3 | 12/30 | 14/30 | 0/30 | **19/30** |

HL = HANDLOOM. HANDLOOM outperforms the baseline and
ablations on all tiers except tier C2. Explanation provided in
Section 5.4.1.

Table 4: Learning From Demos

| Tier | Corr. Trace | Succ. |
|------|-------------|-------|
| B1 | 5/5 | 5/5 |
| B2 | 4/5 | 4/5 |
| B3 | 4/5 | 3/5 |

Corr. Trace = correct trace, or perception
result success. Succ. = perception and
manipulation success.

Table 5: Robot Untangling Experiments (90 total trials)

| | Tier D1 | | Tier D2 | | Tier D3 | |
|---|---|---|---|---|---|---|
| | SGTM 2.0 | HL | SGTM 2.0 | HL | SGTM 2.0 | HL |
| Knot 1 Succ. | 11/15 | **12/15** | 6/15 | **11/15** | 9/15 | **14/15** |
| Knot 2 Succ. | - | - | - | - | 2/15 | **6/15** |
| Verif. Rate | **11/11** | 8/12 | **6/6** | 6/11 | **1/2** | 2/6 |
| Knot 1 Time (min) | 1.1±0.1 | 2.1±0.3 | 3.5±0.7 | 3.9±1.1 | 1.8±0.4 | 2.0±0.4 |
| Knot 2 Time (min) | - | - | - | - | 3.1±1.2 | 7.5±1.6 |
| Verif. Time (min) | 5.7±0.9 | 6.1±1.4 | 6.4±1.8 | 10.1±0.7 | 5.4 | 9.6±1.5 |

HL = HANDLOOM. Across all 3 tiers, HANDLOOM outperforms SGTM 2.0 on
untangling success. SGTM 2.0, however, outperforms HANDLOOM on verification.
Details provided in Section 5.4.2

### 5.4.2 Physical Robot Untangling

For the untangling system based on HANDLOOM, we compare performance against SGTM 2.0 [4], the current state-of-the-art algorithm for untangling long cables, using the same 15-minute timeout on each rollout and the same metrics for comparison. We evaluate HANDLOOM deployed on the ABB YuMi bimanual robot in untangling performance on the following 3 levels of difficulty (Figure 4), where all knots are upward of 10 cm in diameter:

**Tier D1:** Cable with overhand, figure-eight, or overhand honda knot; total crossings $\leq 6$.

**Tier D2:** Cable with bowline, linked overhand, or figure-eight honda knot; total crossings $\in [6, 10)$.

**Tier D3:** Cable with 2 knots (1 each from tiers D1 and D2); total crossings $\in [10, 15)$.

Table 5 shows that our HANDLOOM-based untangling system achieves a higher untangling success rate (29/45) than SGTM 2.0 (19/45) across 3 tiers of difficulty, although SGTM 2.0 is faster. The slower speed of HANDLOOM is attributed to the requirement of a full cable trace for knot detection which is inhibited by the fact that the cable is $3\times$ as long as the width of the workspace, leading to additional reveal moves before performing an action or verifying termination. On the other hand, SGTM 2.0 does not account for the cable exiting the workspace, benefiting speed, but failing to detect off-workspace knots, leading to premature endings of rollouts without fully untangling.

## 6 Limitations and Future Work

HANDLOOM assumes cables are not occluded by large, non-cable objects, are in semi-planar configurations, and are distinguishable from the background, which is one uniform color. Additionally, HANDLOOM is not trained on cables with a wide range of physical (e.g. thickness) and material (elasticity, bending radius) properties, such as loose string or rubbery tubing; and certain empirical experimentation suggests that for significantly thicker ropes and threads that form "kinks", performance of HANDLOOM suffers due to the out-of-distribution nature of these cases. Future work will aim to mitigate these problems and also generalize across more broadly varying backgrounds and properties of cable. Another area of work will involve sampling multiple likely traces from the model and using resultant uncertainty as a signal downstream as a means of mitigating noise.

**Acknowledgments**

This research was performed at the AUTOLAB at UC Berkeley in affiliation with the Berkeley AI Research (BAIR) Lab. The authors were supported in part by donations from Toyota Research Institute.

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

# 7 Appendix

## 7.1 Details on HANDLOOM Methods

### 7.1.1 Over/Undercrossing Predictor

**Model Architecture and Inference:** The binary classification threshold of 0.275 is determined by testing accuracy on a held-out validation set of 75 images on threshold values in the range

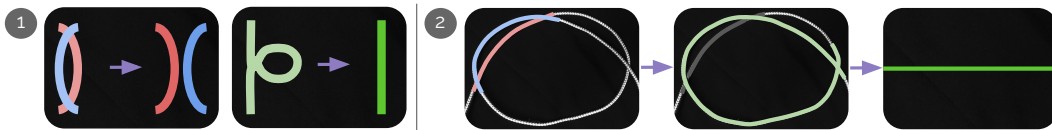

Figure 5: **Reidemeister Moves and Crossing Cancellation**: Left of part 1 depicts Reidemeister Move II. Right of part 1 depicts Reidemeister Move I. Part 2 shows that by algorithmically applying Reidemeister Moves II and I, we can cancel trivial loops, even if they visually appear as knots.

$[0.05, 0.95]$ at intervals of 0.05. Scores $< 0.275$ indicate undercrossing predictions and scores $\geq 0.275$ indicate overcrossing predictions. We output the raw prediction score and a scaled confidence value (0.5 to 1) indicating the classifier's probability.

## 7.2 Details on Robot Untangling using HANDLOOM

### 7.2.1 Knot Definition

Consider a pair of points $p_1$ and $p_2$ on the cable path at time $t$ with $(p_1, p_2 \in \mathcal{C}_t)$. Knot theory strictly operates with closed loops, so to form a loop with the current setup, we construct an imaginary cable segment with no crossings joining $p_1$ to $p_2$ [43]. This imaginary cable segment passes above the manipulation surface to complete the loop between $p_1$ and $p_2$ ("$p_1 \to p_2$ loop"). A knot exists between $p_1$ and $p_2$ at time $t$ if no combination of Reidemeister moves I, II (both shown in Figure 5), and III can simplify the $p_1 \to p_2$ loop to an unknot, i.e. a crossing-free loop. In this paper, we aim to untangle semi-planar knots. For convenience, we define an indicator function $k(s) : [0, 1] \to \{0, 1\}$ which is 1 if the point $\theta(s)$ lies between any such points $p_1$ and $p_2$, and 0 otherwise.

Based on the above knot definition, this objective is to remove all knots, such that $\int k(s)_0^1 = 0$. In other words, the cable, if treated as a closed loop from the endpoints, can be deformed into an unknot. We measure the success rate of the system at removing knots, as well as the time taken to remove these knots.

### 7.2.2 State Definition

We construct line segments between consecutive points on the trace outputted by the learned cable tracer (Section 4.1). Crossings are located at the points of intersection of these line segments. We use the crossing classifier (Section 4.2) to estimate whether these crossings are over/undercrossings. We also implement probabilistic crossing correction with the aim of rectifying classification errors, as we describe in Section 4.2.2.

We denote the sequence of corrected crossings, in the order that they are encountered in the trace, by $\mathcal{X} = (c_1, ..., c_n)$, where $n$ is the total number of crossings and $c_1, ..., c_n$ represent the crossings along the trace.

### 7.2.3 Crossing Cancellation

Crossing cancellation allows for the simplification of cable structure by removing non-essential crossings, shown in Figure 5. It allows the system to filter out some trivial configurations as Reidemeister moves maintain knot equivalence [43]. We cancel all pairs of consecutive crossings ($c_i$, $c_{i+1}$) in $\mathcal{X}$ for some $j$) that meet any of the following conditions:

- *Reidemeister I:* $c_i$ and $c_{i+1}$ are at the same location, or

- *Reidemeister II:* $c_i$ and $c_{i+1}$ are at the same set of locations as $c_j$ and $c_{j+1}$ ($c_j, c_{j+1} \in \mathcal{X}$). Additionally, $c_i$ and $c_{i+1}$ are either both overcrossings or both undercrossings. We also cancel ($c_j, c_{j+1}$) in this case.

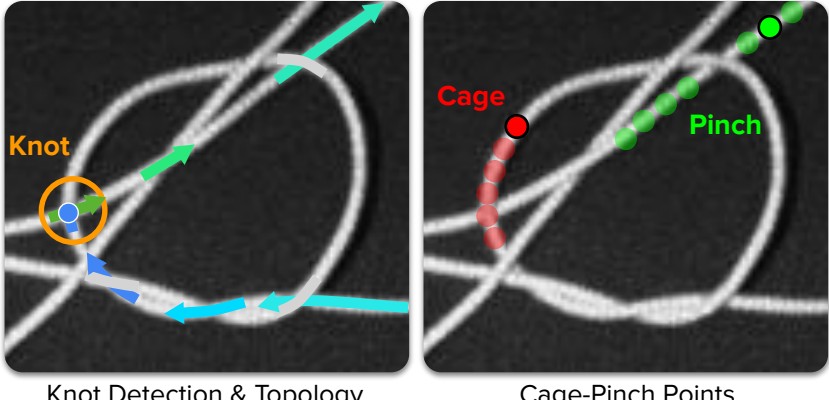

Figure 6: **Knot Detection and Cage Pinch Point Selection**: The left image shows using crossing cancellation rules from knot theory, the knot detection algorithm analytically determines where the knot begins in the cable. The right image shows the survey process for selecting the cap pinch points.

We algorithmically perform alternating Reidemeister moves I and II as described. We iteratively apply this step on the subsequence obtained until there are no such pairs left. We denote the final subsequence, where no more crossings can be canceled, by $\mathcal{X}'$.

### 7.2.4 Knot Detection

We say that a subsequence of $\mathcal{X}'$, $\mathcal{K}_{ij} = (c_i, ..., c_j)$, defines a potential knot if:

- $c_i$ is an undercrossing, and
- $c_j$ is an overcrossing at the same location, and
- at least one intermediate crossing, i.e. crossing in $\mathcal{X}'$ that is not $c_i$ or $c_j$, is an overcrossing.

The first invariant is a result of the fact that all overcrossings preceding the first undercrossing (as seen from an endpoint) are removable. We can derive this by connecting both endpoints from above via an imaginary cable (as in Section 7.2.1): all such overcrossings can be removed by manipulating the loop formed. The second invariant results from the fact that a cable cannot be knotted without a closed loop of crossings. The third and final invariant can be obtained by noting that a configuration where all intermediate crossings are undercrossings reduces to the unknot via the application of the 3 Reidemeister moves. Therefore, for a knot to exist, it must have at least one intermediate overcrossing.

Notably, these conditions are necessary, but not sufficient, to identify knots. However, they improve the likelihood of bypassing trivial configurations and detecting knots. This increases the system's efficiency by enabling it to focus its actions on potential knots.

### 7.2.5 Algorithmic Cage-Pinch Point Detection

As per the definition introduced in Section 7.2.4, given knot $\mathcal{K}_{ij} = (c_i, ..., c_j)$, $c_i$ and $c_j$ define the segments that encompass the knot where $c_i$ is an undercrossing and $c_j$ is an overcrossing for the same crossing. The pinch point is located on the overcrossing cable segment, intended to increase space for the section of cable and endpoint being pulled through. The cage point is located on the undercrossing cable segment. To determine the pinch point, we search from crossing $c_{u1}$ to crossing $c_{u2}$. $c_{u1}$ is the previous undercrossing in the knot closest in the trace to $j$. $u2 > j$ and $c_{u2}$ is the next undercrossing after the knot. We search in this region and select the most graspable region to pinch at, where graspability $(G)$ is defined by the number of pixels that correspond to a cable within a given crop and a requirement of sufficient distance from all crossings $c_i$. To determine the cage point, we search from crossing $c_i$ to $c_k$ where $i < k < j$ and $c_k$ is the next undercrossing in the knot closest in the trace to $c_i$. We similarly select the most graspable point. If no points in the search

space for either the cage or pinch point are graspable, meaning $G < \mathcal{T}$ where $\mathcal{T}$ is an experimentally derived threshold value, we continue to step along the trace from $c_{u2}$ for pinch and from $c_k$ for cage until $G \geq \mathcal{T}$. This search process is shown in Figure 6.

### 7.2.6 Manipulation Primitives

We use the same primitives as in SGTM 2.0 (Sliding and Grasping for Tangle Manipulation 2.0) [4] to implement HANDLOOM as shown in Figure 7 for untangling long cables. We add a *perturbation* move.

**Cage-Pinch Dilation:** We use cage-pinch grippers as in Viswanath et al. [3]. We have one gripper cage grasp the cable, allowing the cable to slide between the gripper fingers but not slip out. The other gripper pinch grasps the cable, holding the cable firmly in place. This is crucial for preventing knots in series from colliding and tightening during untangling. The *partial* version of this move introduced by Shivakumar et al. [4] separates the grippers to a small, fixed distance of 5 cm.

**Reveal Moves:** First, we detect endpoints using a Mask R-CNN object detection model. If both endpoints are visible, the robot performs an *Endpoint Separation Move* by grasping at the two endpoints and then pulling them apart and upwards, away from the workspace, allowing gravity to help remove loops before placing the cable back on the workspace. If both endpoints are not visible, the robot performs an *Exposure Move*. This is when it pulls in cable segments exiting the workspace. Building on prior work, we add a focus on where this move is applied. While tracing, if we detect the trace hits the edge, we perform an exposure move at the point where the trace exits the image.

**Perturbation Move:** If an endpoint or the cable segment near an endpoint has distracting cable segments nearby, making it difficult for the analytic tracer to trace, we perturb it by grasping it and translating in the x-y plane by uniformly random displacement in a $10\text{cm} \times 10\text{cm}$ square in order to separate it from slack.

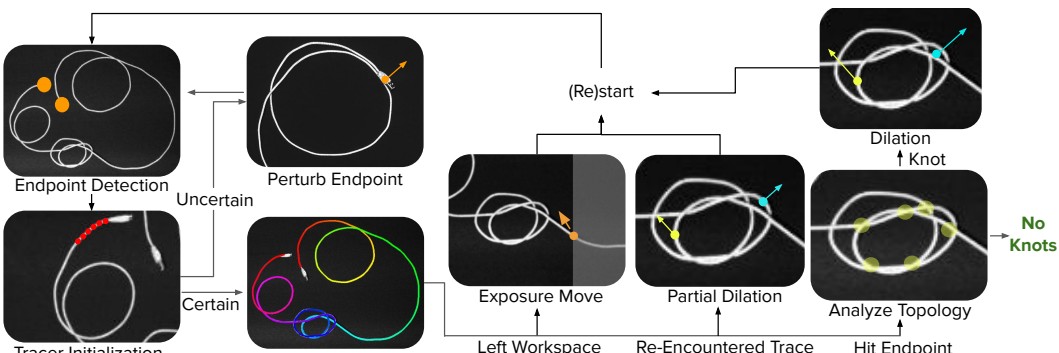

Figure 7: **Untangling Algorithm with HANDLOOM**: We first detect the endpoints and initialize the tracer with start points. If we are not able to obtain start points, we perturb the endpoint and try again. Next, we trace. While tracing, if the cable exits the workspace, we pull the cable towards the center of the workspace. If the tracer gets confused and begins retracing a knot region, we perform a partial cage-pinch dilation that will loosen the knot, intended to make the configuration easier to trace on the next iteration. If the trace is able to successfully complete, we analyze the topology. If there are no knots, we are done. If there are knots, we perform a cage-pinch dilation and return to the first step.

### 7.2.7 Cable Untangling System

Combining HANDLOOM and the manipulation primitives from Section 7.2.6, the cable untangling algorithm works as follows: First, detect endpoints and initialize the learned tracer with 6 steps of the analytic tracer. If HANDLOOM is unable to get these initialization points, perturb the endpoint from which we are tracing and return to the endpoint detect step. Otherwise, during tracing, if the cable leaves the workspace, perform an exposure move. If the trace fails and begins retracing itself, which can happen in denser knots, perform a partial cage-pinch dilation as in [4]. If the trace completes and reaches the other endpoint, analyze the topology. If knots are present, determine the

| Cable Reference | TR | 1 | 2 | 3 | 4 | 5 | Avg. |
|---|---|---|---|---|---|---|---|
| Tracing Success Rate | 6/8 | 7/8 | 8/8 | 7/8 | 6/8 | 6/8 | 40/48=83% |
| Failures | (I) 2 | (I) 1 | | (II) 1 | (I) 1, (III) 1 | (II) 1, (III) 1 | |

cage-pinch points for it, apply a cage-pinch dilation move to them, and repeat the pipeline. If no knots are present, the cable is considered to be untangled. The entire system is depicted in Figure 7.

## 7.3 Details on Knot Tying Experiments

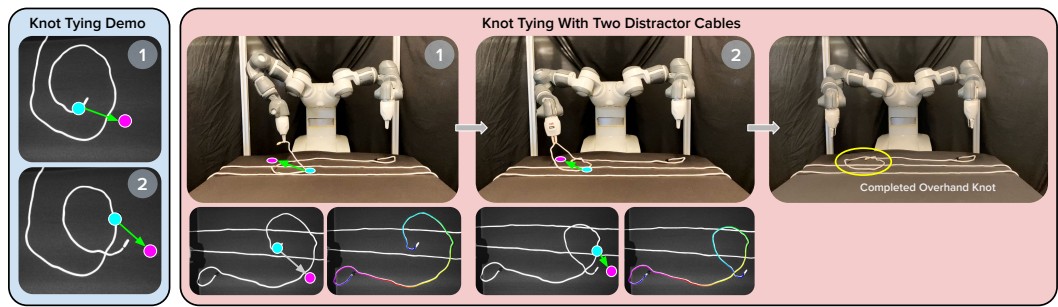

Figure 8: **Using HANDLOOM for Learning from Demos:** The left (blue panel) displays the single human demonstration, indicating the pick and place points for tying an overhand knot. The right (pink panel) shows this demonstration successfully applied to the cable in a different configuration with 2 other distractor cables in the scene. The first step of the demonstration is achieved through an arc length relative action while the second step is achieved through a crossing relative action.

When performing state-based imitation, each of the pick and place points $p_i$ from the demonstration is parameterized in the following way: 1) find the point along the trace, $T$, closest to the chosen point $\hat{p}_i$ with index $j$ in $T$, 2) find the displacement $d_i = p_i - \hat{p}_i$ in the local trace-aligned coordinate system of $\hat{p}_i$, 3) in memory, for point $p_i$, store $d_i$, arc length of $\hat{p}_i$ ($\sum_{x=1}^{j} T_x - T_{x-1}$), and the index value of the crossing in the list of crossings just before $\hat{p}_i$.

When rolling out a policy using this demonstration, there are two ways to do so: 1) relative to the arc length along the cable, or 2) relative to the fraction of the arc length between the 2 crossing indices. The way to do so is to find the point on the cable with the same arc length as $\hat{p}_i$ from the demo or the fractional arc length between the same 2 crossing indices, depending on the type of demonstration. Then, apply $d_i$ in the correct trace-aligned coordinate system. An example demonstration is shown in Figure 8.

## 7.4 Experiments Failure Mode Analysis

### 7.4.1 Using HANDLOOM for Tracing Cables Unseen During Training

(1) Retraces previously traced cable (went in a loop).

(2) Missteps onto a parallel cable.

(3) Skips a loop.

Figure 9 shows the cables tested on. The most common failure mode is (I), retracing previously traced cable. This is commonly observed in cases with near parallel segments or in dense loop areas within a knot.

### 7.4.2 Using HANDLOOM for Cable Inspection in Multi-Cable Settings

(I) Misstep in the trace, i.e. the trace did not reach any adapter.

(II) The trace reaches the wrong adapter.

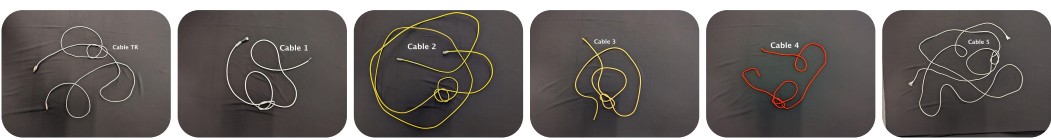

Figure 9: **Cables for Tracing Unseen Cables Experiment**

Table 7: Multi-Cable Tracing Results

|  | Analytic | Learned |
|---|---|---|
| Tier A1 | 3/30 | **27/30** |
| Tier A2 | 2/30 | **23/30** |
| Tier A3 | 1/30 | **23/30** |
| Failures | (I) 3, (II) 45, (III) 36 | (I) 14, (II) 1, (III) 2 |

(III) The trace reaches the correct adapter but is an incorrect trace.

The most common failure mode for the learned tracer, especially in Tier A3, is (I). One reason for such failures is the presence of multiple twists along the cable path (particularly in Tier A3 setups, which contain more complex inter-cable knot configurations). The tracer is also prone to deviating from the correct path on encountering parallel cable segments. In Tier A2, we observe two instances of failure mode (III), where the trace was almost entirely correct in that it reached the correct adapter but skipped a section of the cable.

The most common failure modes across all tiers for the analytic tracer are (II) and (III). The analytic tracer particularly struggles in regions of close parallel cable segments and twists. As a result of the scoring metric, 87 of the 90 paths that we test reach an adapter; however, 45/90 paths did not reach the correct adapter. Even for traces that reach the correct adapter, the trace is incorrect, jumping to other cables and skipping sections of the true cable path.

### 7.4.3 Using HANDLOOM for Physical Robot Knot Tying from Demonstrations

(1) Trace missteps onto a parallel cable.

(2) Cable shifted during manipulation, not resulting in a knot at the end.

Failure mode (1) occurs when the distractor cable creates near parallel sections to the cable of interest for knot tying, causing the trace to misstep. Failure mode (2) occurs when the manipulation sometimes slightly perturbs the rest of the cable's position while moving one point of the cable, causing the end configuration to not be a knot, as intended.

### 7.4.4 Using HANDLOOM for Knot Detection

(A) The system fails to detect a knot that is present—a false negative.

(B) The system detects a knot where there is no knot present—a false positive.

(C) The tracer retraces previously traced regions of cable.

(D) The crossing classification and correction schemes fail to infer the correct cable topology.

(E) The knot detection algorithm does not fully isolate the knot, also getting surrounding trivial loops.

(F) The trace skips a section of the true cable path.

(G) The trace is incorrect in regions containing a series of close parallel crossings.

Table 8: Learning From Demos

|  | Succ. Rate | Failures |
|---|---|---|
| Tier B1 | 5/5 | - |
| Tier B2 | 4/5 | (1) 1 |
| Tier B3 | 4/5 | (1) 1, (2) 1 |

Table 9: HANDLOOM Experiments

|  | SGTM 2.0 | HANDLOOM (-LT) | HANDLOOM (-CC) | HANDLOOM |
|---|---|---|---|---|
| Tier C1 | 2/30 | 14/30 | 20/30 | **24/30** |
| Tier C2 | **28/30** | 8/30 | 21/30 | 26/30 |
| Tier C3 | 12/30 | 14/30 | 0/30 | **19/30** |
| Failures | (A) 30, (B) 18 | (D) 11, (F) 7 (G) 24, (H) 11 | (B) 38, (C) 5, (E) 6 | (B) 11, (D) 8 (F) 1 |

Table 10: HANDLOOM and Physical Robot Experiments (90 total trials)

|  | Tier D1 | | Tier D2 | | Tier D3 | |
|---|---|---|---|---|---|---|
|  | SGTM 2.0 | HANDLOOM | SGTM 2.0 | HANDLOOM | SGTM 2.0 | HANDLOOM |
| Knot 1 Succ. | 11/15 | **12/15** | 6/15 | **11/15** | 9/15 | **14/15** |
| Knot 2 Succ. | - | - | - | - | 2/15 | **6/15** |
| Verif. Rate | **11/11** | 8/12 | **6/6** | 6/11 | **1/2** | 2/6 |
| Knot 1 Time (min) | 1.1±0.1 | 2.1±0.3 | 3.5±0.7 | 3.9±1.1 | 1.8±0.4 | 2.0±0.4 |
| Knot 2 Time (min) | - | - | - | - | 3.1±1.2 | 7.5±1.6 |
| Verif. Time (min) | 5.7±0.9 | 6.1±1.4 | 6.4±1.8 | 10.1±0.7 | 5.4 | 9.6±1.5 |
| Failures | (7) 4 (1) 2, (2) 1 | (1) 2, (2) 1 | (1) 3, (5) 6 (1) 3, (5) 6 | (2) 2, (4) 1 (5) 1 | (1) 3, (2) 3, (5) 3 (6) 2, (7) 2 | (1) 2, (2) 3 (3) 1, (6) 3 |

(H) The tracer takes an incorrect turn, jumping to another cable segment.

For SGTM 2.0, the most common failure modes are (A) and (B), where it misses knots or incorrectly identifies knots when they are out of distribution. For HANDLOOM (-LT), the most common failure modes are (F), (G), and (H). All 3 failures are trace-related and result in knots going undetected or being incorrectly detected. For HANDLOOM (-CC), the most common failure modes are (B) and (E). This is because HANDLOOM (-CC) is unable to distinguish between trivial loops and knots without the crossing cancellation scheme. By the same token, HANDLOOM (-CC) is also unable to fully isolate a knot from surrounding trivial loops. For HANDLOOM, the most common failure mode is (B). However, this is a derivative of failure mode (D), which is present in HANDLOOM (-LT), HANDLOOM (-CC), and HANDLOOM. Crossing classification is a common failure mode across all systems and is a bottleneck for accurate knot detection. In line with this observation, we hope to dig deeper into accurate crossing classification in future work.

### 7.4.5 Using HANDLOOM for Physical Robot Untangling

(1) Incorrect actions create a complex knot.

(2) The system misses a grasp on tight knots.

(3) The cable falls off the workspace.

(4) The cable drapes on the robot, creating an irrecoverable configuration.

(5) False termination.

(6) Manipulation failure.

(7) Timeout.

The main failure modes in HANDLOOM are (1), (2), and (6). Due to incorrect cable topology estimates, failure mode (1) occurs: a bad action causes the cable to fall into complex, irrecoverable states. Additionally, due to the limitations of the cage-pinch dilation and endpoint separation moves, knots sometimes get tighter during the process of untangling. While the perception system is still able to perceive the knot and select correct grasp points, the robot grippers bump the tight knot, moving the entire knot and causing missed grasps (2). Lastly, we experience manipulation failures while attempting some grasps as the YuMi has a conservative controller (6). We hope to resolve these hardware issues in future work.

The main failure modes in SGTM 2.0 are (5) and (7). Perception experiments indicate that SGTM 2.0 has both false positives and false negatives for cable configurations that are out of distribution. (5) occurs when out-of-distribution knots go undetected. (7) occurs when trivial loops are identified as knots, preventing the algorithm from terminating.

