# OpenReview forum: "HANDLOOM: Learned Tracing of One-Dimensional Objects for Inspection and Manipulation"
_robot-learning.org/CoRL/2023/Conference — CoRL 2023 Oral_

### Official Review · Reviewer_KUAt · 2023-07-09

**Confidence:** 5
**Originality:** Very Good
**Technical Quality:** Excellent
**Clarity Of Presentation:** Very Good
**Impact:** 4

**Recommendation:**

Strong Accept: I recommend accepting the paper and will argue for my recommendation even if other reviewers hold a different opinion.

**Review:**

This paper presents a novel method and very impressive results in the task of detecting one-dimensional objects.
Performed evaluation is very thorough and shows the high accuracy of the proposed approach and high applicability potential in downstream tasks. The presented approach seems to be a significant contribution to the field of one-dimensional object tracing.


Strengths

1. Interesting approach.
1. Extensive real-world evaluation.
2. Public dataset and code.
3. High success rate on very challenging benchmarks - very impressive.
4. Presented application in downstream tasks.
5. Clarity of the presentation.

Weaknesses

1. In terms of tracing one-dimensional objects many important related works are not mentioned [1, 2, 3].

2. Do you expect your method to be robust to some segmentation artifacts that are common for not plain backgrounds?

3. Running time of the method itself is not mentioned (only the execution of the downstream tasks). How long does it take to trace a 3m cable?


[1] A. Caporali, K. Galassi, R. Zanella and G. Palli, "FASTDLO: Fast Deformable Linear Objects Instance Segmentation," in IEEE Robotics and Automation Letters, vol. 7, no. 4, pp. 9075-9082, Oct. 2022, doi: 10.1109/LRA.2022.3189791.


[2] A. Caporali, R. Zanella, D. D. Greogrio and G. Palli, "Ariadne+: Deep Learning--Based Augmented Framework for the Instance Segmentation of Wires," in IEEE Transactions on Industrial Informatics, vol. 18, no. 12, pp. 8607-8617, Dec. 2022, doi: 10.1109/TII.2022.3154477.

[3] P. Kicki, A. Szymko and K. Walas, "DLOFTBs – Fast Tracking of Deformable Linear Objects with B-splines," 2023 IEEE International Conference on Robotics and Automation (ICRA), London, United Kingdom, 2023, pp. 7104-7110, doi: 10.1109/ICRA48891.2023.10160437.

**Quality Of The Limitations Section:**

Limitations are addressed clearly

**Questions For Rebuttal:**

1. What about non-semi-panar cases? What are the limitations of this method in these cases? Is it only a matter of relevant data, or there are more substantial problems with it?

2. Do you expect your method to be robust to some segmentation artifacts that are common for not plain backgrounds?

3. Running time of the method itself is not mentioned (only execution of the downstream tasks). How long does it take to trace a 3m cable?

4. Why grayscale? Colors could help in finding the true continuation of the DLO.

5. What is the impact of lighting? Have you tested and trained in the same lighting conditions?

**Robotics Focus:**

Sufficient demonstration on hardware

**Summary Of Paper:**

This paper proposes a novel learning-based method of tracing one-dimensional objects on grayscale images. This approach works in an autoregressive manner and traverses the one-dimensional object by identifying the points on the cable step by step using local image crops. The proposed approach allows for tracing very long objects in very challenging scenarios despite intersections with other identical objects and itself. The authors presented also the application of the proposed method to downstream tasks, such as cable untangling.

**Summary Of Recommendation:**

Due to the high quality of the manuscript, proposed method, and performed experiments I'm convinced that this paper should be accepted.

---

### Official Review · Reviewer_XRHn · 2023-07-12

**Confidence:** 4
**Originality:** Excellent
**Technical Quality:** Excellent
**Clarity Of Presentation:** Excellent
**Impact:** 4

**Recommendation:**

Strong Accept: I recommend accepting the paper and will argue for my recommendation even if other reviewers hold a different opinion.

**Review:**

This paper is well presented, well structured and well evaluated.
One could regret that the methodology is presented too fast, but this is clearly a result of the page limit for the conference.

The overall approach is interesting and original. The performances are very convincing and the potential for applications very interesting.

Minor typos could be fix to improve the paper. Specifically, figure 3 refers to a bottom row but there seems to be a single row in the figure.


**Quality Of The Limitations Section:**

Limitations are addressed clearly

**Questions For Rebuttal:**

As a minor question that could be discussed in the paper (and in the rebuttal): I did not get what the over/under classifier was using to take its decision. Does it rely on shadows from one cable on the other ? This question is related to the generalizability of this task to different applications where the traces to untangle are not the physical representation of a cable, but a hand drawing, or a trajectory in a tracking application.

**Robotics Focus:**

Sufficient demonstration on hardware

**Summary Of Paper:**

This paper presents a deep-learning based scene-understanding method for entangled cables from 2D images. The methods is trained with images of white cables on a black background and learns first to segment and trace the cables from the background,  and then to detect line crossing and classify them as crossing-over or crossing-under. The tracing part takes an image with a part of the cable known and predict where the cable will continue.

The method is trained on a mix of real and simulated examples and then tested on real images, with a diversity of cables type showing the generalization of the method.

Finally, the method is used with a dual-arm robot manipulator to disentangle cables on a table.

The approach is evaluated quantitatively and  compared to the state of the art.

The additional material includes dataset and will include a git repo.

**Summary Of Recommendation:**

A very enjoyable paper which not only present an interesting concept but also suggest new ideas for other applications.

---

### Official Review · Reviewer_bwb5 · 2023-07-21

**Confidence:** 3
**Originality:** Good
**Technical Quality:** Good
**Clarity Of Presentation:** Excellent
**Impact:** 3

**Recommendation:**

Weak Accept: I recommend accepting the paper, but will not argue for my recommendation if the majority of other reviewers have a different opinion.

**Review:**

Pros:
- I really appreciate the transparency of the paragraph on "Workspace and Assumptions". It so clearly lays out limiting assumptions about the input images instead of obfuscating them in experimental results. It really helps to understand where this method can be applied.
- Very clearly organized, easy to follow, and enjoyable to read.
- Clear task formulation and a reasonable method. It is easy to see how and why the method solves the task, and how it works.
- I am not an expert on cable tracing or knot tying, but the results really look quite good. The method traces cables correctly ~80% of the time, and each example consists of a lot of crossings in a pretty big mess.
- The experiments are very extensive, showing multiple downstream uses. I appreciate the dedication. In learning untangling, the proposed method outperforms prior state of the art.

Cons:
- Some of the motivation is a little far fetched and doesn't come across as intellectually honest.
- The semi-planar assumption limits applicability in practical untangling scenarios,
where cables would form 3D structures. It's not really a flaw of the paper though: the paper clearly states this assumption, which helps.


Minor comments and errors:
- The first paragraph in introduction is a bit of a stretch and a tad dramatic. I am sure there are more realistic solutions to messy cabling compared to deploying a robot. None of the works cited to substantiate this [1,2,3,4] actually study safety issues of messy cabling. [1] is a survey of methods, [2,3] are on surgical robotics, [4] tries to tie knots (not untie anything). I would remove this paragraph - it motivates the work with a shaky premise. This research has academic and intellectual value even without a clear application in mind. If the authors do wish to motivate the work practically, there are more realistic applications. For example, robots in automated testing for quality assurance might be applied to plug cables into devices to check that the connectors work.
- Line 111 has a typo.
- Figure 3 caption mentions 2 rows, but there is only 1 row. The second row sounds interesting, I would like to see it. But right now I don't understand the image.

**Quality Of The Limitations Section:**

Limitations are addressed clearly

**Questions For Rebuttal:**

- Are more comparisons to prior work possible, especially on the cable tracing and knot detection tasks? I am not very familiar with cable manipulation so I am not sure, but a response on this would help.

**Robotics Focus:**

Sufficient demonstration on hardware

**Summary Of Paper:**

The paper proposes a method to estimate the state of long and thin cables in semi-planar configurations from RGBD observations. The method consists of a cable tracer model and a crossing classifier. The tracer auto-regressively predicts the next coordinate on a cable, conditioned on the history of prior keypoints. The crossing classifier takes as input a small image crop centered on the crossing of two traced cables, and predicts which cable is on top. The models are trained on synthetic images of cables synthesized in Blender using curve-generation heuristics to produce diverse and interesting cable crossings. A small number of hand-labeled real images are included as well.

The paper includes experiments on 3 main tasks: tracing unseen cables, learning knot tying from demonstrations, and cable untangling. The paper shows compelling performance, including enabling better learning of untangling behavior compared to prior state of the art.

**Summary Of Recommendation:**

Overall I enjoyed the paper, learned an interesting method, and was sufficiently convinced that the proposed method works well at solving the stated task. I appreciate the transparency of the assumptions. My main concern is around the narrow applicability of the method (in particular the semi-planar assumption), and I was not sure if there is sufficient comparison to prior work (I honestly don't know).

---

### Author Response · Authors · 2023-08-12
**Thank you for all the insightful comments and feedback!**

Thank you to all the reviewers for your positive feedback and suggestions on improvements (Summary: 1 weak accept, 2 strong accepts)  We have revised the paper accordingly (attached in the reviewer responses), with changes in red, and address the individual comments in the attached responses. We look forward to further discussion.

---

### Decision · Program_Chairs · 2023-08-30

**Decision:**

Accept (Oral)

**Comment:**

This paper presents a method for estimating the configuration of deformable 1D objects (i.e. cables, ropes) using visual data. The method "traces" the object in the image to estimate the configuration, especially estimating whether crossings are "over" or "under". Several impressive results are shown with very tangled cables being correctly traced.

Strengths:
- The approach is extensively evaluated in the real-world, achieving a high success rate
- The dataset and code are open
- The paper is generally clear and understandable

Weaknesses
- Some related work is overlooked and should be included (see reviewer comments)
- The method relies on effective segmentation. Though segmentation is not within the scope of the work, the authors could consider incorporating a method to deal with segmentation artifacts that will be inevitable.
- The results should present run-time statistics

Overall, the rebuttal did not significantly change the reviewers' opinions.